# ON THE FISHER GEOMETRY OF DIFFUSION MODELS' LATENT SPACE

**Maria Esteban-Casadevall** [*]
University of Amsterdam

**Rafał Karczewski** [†]
Aalto University

**Alison Pouplin**[†]
Bayer AG

**Søren Hauberg**
Technical University of Denmark

**Erik J. Bekkers**
University of Amsterdam

## ABSTRACT

In this paper, we advance the understanding of the geometry of the latent space of diffusion models through the lens of information geometry. In particular, we consider the latent variable $\boldsymbol{\theta} = (\boldsymbol{x}_t, t)$ that indexes a family of denoising distributions $p(\boldsymbol{x}_0|\boldsymbol{x}_t)$ across all noise levels, which naturally defines a statistical manifold endowed with the Fisher-Rao metric. We study how the choice of noise schedule affects the geometry of this manifold, and how the geometry evolves along the denoising process. We show that, under mild assumptions, the statistical manifolds induced by different noise schedules are isometric, and we demonstrate a link between the curvature and phase-transition-like behavior.

## 1 INTRODUCTION

Diffusion models gradually corrupt a data sample $\boldsymbol{x}_0 \in \mathbb{R}^D$ into Gaussian noise $\boldsymbol{x}_T$ through $T$ forward steps. By learning to reverse this process, they have achieved great success in generative modeling tasks (Song et al., 2022; Yang et al., 2023). Recent work has focused on understanding the latent structure of these models to gain further insights into its inner workings (Park et al., 2023; Lobashev et al., 2025).

In this work, we build on the framework introduced by (Karczewski et al., 2026), which defines the latent space as the spacetime continuum $(\boldsymbol{x}_t, t)$ for $t \in (0, T]$ and endows the family of denoising distributions $p(\boldsymbol{x}_0 \mid \boldsymbol{x}_t)$ with the Fisher metric. (Karczewski et al., 2026) shows that this construction defines an *exponential* statistical manifold with tractable geodesics. We show that, under reasonable assumptions, all variance-preserving noise schedules induce the same geometry, but give rise to noise-dependent distortions when studied through the *fixed* spacetime coordinate chart. We demonstrate that, in the case of a Gaussian mixture prior, this manifold contains regions of negative curvature, and exhibits a notion of *phase-transition*, where the Fisher metric changes abruptly.

## 2 BACKGROUND

**Diffusion models.** Let $\mathcal{X} \subseteq \mathbb{R}^D$ be the data space, and let $h$ be the data distribution on $\mathcal{X}$. For $\boldsymbol{x}_0 \in \mathcal{X}$ we consider the forward diffusion process

$$p(\boldsymbol{x}_t \mid \boldsymbol{x}_0) = \mathcal{N}(\boldsymbol{x}_t \mid \alpha_t \boldsymbol{x}_0, \sigma_t^2 I), \tag{1}$$

which gradually transforms $h$ into pure noise $p_T \approx \mathcal{N}(\boldsymbol{0}, \sigma_T^2 I)$ at time $T$, where $\alpha_t$ and $\sigma_t$ define the forward drift $f_t$ and diffusion $g_t$. We will be working with the variance-preserving (VP) framework where $\alpha_t^2 + \sigma_t^2 = 1$. As defined in (Anderson, 1982), there exist a stochastic differential equation (SDE) that is able to reverse this process. In practice, a deterministic Probability Flow ODE (PF-ODE) with the same marginals is used

$$\text{PF ODE:} \quad d\boldsymbol{x} = \Big(f_t \boldsymbol{x} - \tfrac{1}{2} g_t^2 \nabla \log p_t(\boldsymbol{x})\Big) dt, \quad \boldsymbol{x}_T \sim p_T,$$

---

[*]m.estebancasadevall@uva.nl
[†]Equal contribution

where $p_t$ is the marginal distribution of the forward process at time $t$ (Song et al., 2021).

**Information geometry.** Information geometry is the study of statistical manifolds (Amari and Nagaoka, 2000). These manifolds, whose points are probability distributions $p(\cdot \mid \boldsymbol{\theta})$ parametrized by $\boldsymbol{\theta}$, are equipped with a Riemannian metric: the Fisher-Rao metric (Amari, 2016), which is defined as

$$g^{\mathrm{FR}}(\boldsymbol{\theta}) = \mathbb{E}_{\boldsymbol{x} \sim p(\boldsymbol{x}|\boldsymbol{\theta})} \left[ \nabla_{\boldsymbol{\theta}} \log p(\boldsymbol{x} \mid \boldsymbol{\theta}) \nabla_{\boldsymbol{\theta}} \log p(\boldsymbol{x} \mid \boldsymbol{\theta})^{\top} \right] \in \mathbb{R}^{\dim(\Theta) \times \dim(\Theta)}. \qquad (2)$$

## 3 STATISTICAL MANIFOLD OF DENOISING DISTRIBUTIONS

### 3.1 CONSTRUCTION OF THE SPACETIME MANIFOLD

In this section, we introduce the statistical manifold associated with the denoising distributions of a diffusion process, and study how its geometry depends on the choice of noise schedule.

Following (Karczewski et al., 2026), we consider a latent coordinate $\boldsymbol{\theta} = (\boldsymbol{x}_t, t)$, and its corresponding denoising posterior $p(\boldsymbol{x}_0|\boldsymbol{\theta})$. The family $\{ p(\cdot \mid \boldsymbol{\theta}) \}_{\boldsymbol{\theta} \in \Theta}$ forms a statistical manifold $\mathcal{P}$, equipped with the Fisher–Rao metric $g^{\mathrm{FR}}$. A diffusion process then provides a particular *coordinate representation* of a submanifold of $\mathcal{P}$, constructed as follows (see Figure 1).

**Lemma 1** (Denoising manifold induced by a diffusion process). *Fix a VP noise schedule* $\{\alpha_t, \sigma_t\}_{t \in (0,T]}$. *Consider the Gaussian forward diffusion process* $p(\boldsymbol{x}_t \mid \boldsymbol{x}_0) = \mathcal{N}(\boldsymbol{x}_t \mid \alpha_t \boldsymbol{x}_0, \sigma_t^2 I)$ *for* $t \in (0, T]$ *and the parameter space* $\Theta := \mathbb{R}^d \times (0, T]$. *Define the map*

$$\iota_\alpha : \Theta \to \mathcal{P}, \qquad \iota_\alpha(\boldsymbol{x}_t, t) := p_\alpha(\boldsymbol{x}_0 \mid \boldsymbol{x}_t),$$

*which assigns to each noisy observation* $(\boldsymbol{x}_t, t)$ *the corresponding conditional denoising distribution. Then* $\iota_\alpha$ *is a well-defined coordinate map whose image*

$$\mathcal{M}_\alpha := \iota_\alpha(\Theta) \subset \mathcal{P}$$

*is an embedded submanifold of* $\mathcal{P}$. *Its metric is the Fisher metric restricted to this submanifold.*

We are interested in the following question:

> *Given two different VP noise schedules* $\{\alpha_i(t), \sigma_i(t)\}_{t \in (0,T]}$ *for* $i \in \{1, 2\}$, *how are the geometries of the corresponding denoising manifolds* $(\mathcal{M}_{\alpha_1}, g_1^{FR})$ *and* $(\mathcal{M}_{\alpha_2}, g_2^{FR})$ *related to each other?*

**Definition 1** (Proper time change). A function $\phi : [0, T] \to [0, T]$ is called a *proper time change* if it is smooth, strictly increasing, and boundary-preserving, i.e., $\phi(0) = 0$ and $\phi(T) = T$.

Note that a proper time change exists between any two schedules with matching minimum and maximum noise levels, and monotonic $\alpha$ and $\sigma$. These are properties that most noise schedules satisfy in practice.

**Proposition 1** (Isometry via proper time change). *Let* $\alpha_i, \sigma_i : [0, T] \to \mathbb{R}$, $i = 1, 2$, *be two VP noise schedules associated with a Gaussian forward diffusion processes, and let* $(\mathcal{M}_{\alpha_i}, g_i^{FR})$ *denote the corresponding denoising manifolds. Then the following statements are equivalent*

  (i) *The Riemannian manifolds* $(\mathcal{M}_{\alpha_1}, g_1^{FR})$ *and* $(\mathcal{M}_{\alpha_2}, g_2^{FR})$ *are isometric.*

 (ii) *There exists a proper time change* $\phi : [0, T] \to [0, T]$ *relating the two noise schedules.*

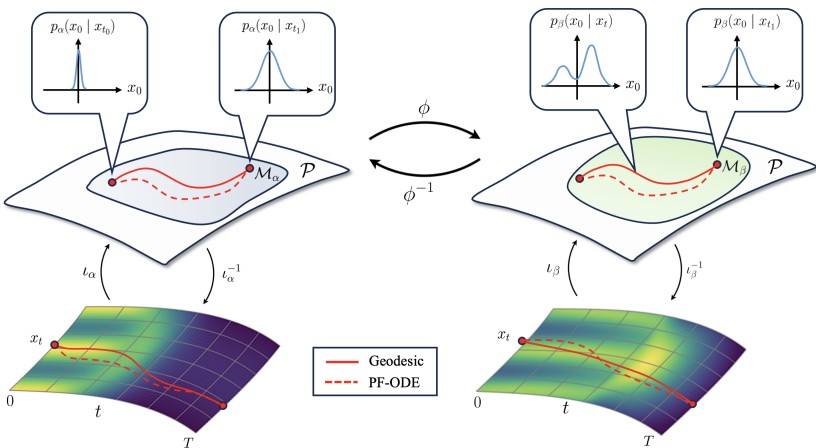

Figure 1: Denoising manifold and spacetime manifold for two different noise schedules, $\alpha$ and $\beta$.

In other words, the geometry of denoising manifold induced by different noise schedules is equivalent up to a reparametrization of time. This provides a geometric interpretation of a well-known concept in diffusion models: proper time changes squeezes and stretches the time axis but does not fundamentally change the diffusion process (Stancevic et al., 2025; Kingma et al., 2021) (See Appendix 5.5 for an interpretation in terms of the forward SDE).

However, despite this equivalence, the empirical performance of diffusion models is highly sensitive to the choice of noise schedule (Chen, 2023; Song et al., 2022; Aranguri et al., 2025). We shed light on this phenomenon by analyzing the geometry through a *fixed* coordinate chart.

**Definition 2** (Spacetime manifold). The *spacetime manifold* associated with a noise schedule $\alpha$ is defined as $(\Theta, g_\alpha)$, where $\Theta := \mathbb{R}^d \times (0, T]$, and the metric $g_\alpha$ is the pullback of the Fisher–Rao metric via $\iota_\alpha : \Theta \to \mathcal{M}_\alpha \subset \mathcal{P}$, i.e $g_\alpha := \iota_\alpha^* g^{\mathrm{FR}}$, and $g^{\mathrm{FR}}$ is the Fisher metric in $\mathcal{M}_\alpha$.

The coordinates $(\boldsymbol{x}_t, t) \in \Theta$ correspond to the observed noisy samples and diffusion time. Intuitively, although the geometry of the statistical manifold is independent of noise schedule (Proposition 1), this invariance is broken in practice when time is not reparametrized accordingly, making numerical ODE solvers struggle through regions of high curvature.

A natural question arises: How *efficiently* does the reverse PF-ODE traverse the statistical manifold, and how does this differ across noise schedules? To get some insights, we look into the Fisher speed of PF-ODE flows under different noise schedules for a Gaussian mixture prior, as shown in Figure 2. Intuitively, the Fisher speed measures how quickly information is gained through the denoising trajectory (see Appendix 5.1 for a formal definition). As expected, the speed of the PF-ODE is not constant and increases as we get close to the data distribution. Large variations in the Fisher speed of trajectories may lead to suboptimal sampling curves. In addition, moving at constant Fisher speed may be beneficial for sampling, as each time step would cover an equal statistical difference (see (Stancevic et al., 2025; Zhang et al., 2025)). As future work, it would be interesting to construct a discretizations scheme that ensures each time step yields an equal gain in information, as recently explored by (Raya et al., 2026).

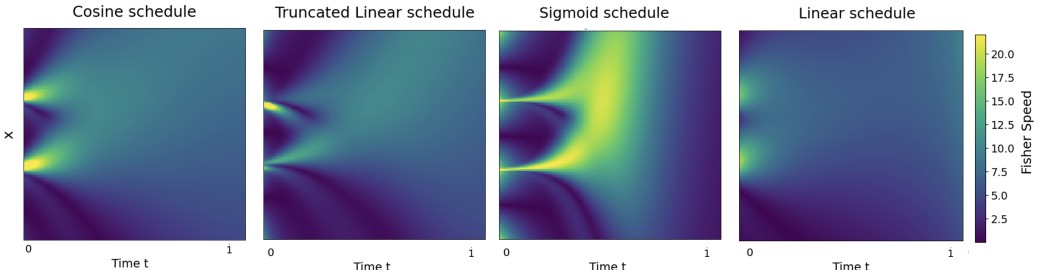

Figure 2: Fisher speed of the PF-ODE under different noise schedules. The data distribution is a Gaussian mixture of three low variance components centered at $-1$, $0$ and $1$. See Appendix 5.7 for the definition of each noise schedule.

## 3.2 THE GEOMETRY OF SPACETIME

The aim of this section is to study how the geometry of the *spacetime manifold* (Definition 2) changes along the denoising process. For simplicity, we will assume a Gaussian mixture prior.

**Definition 3.** Let $h(\boldsymbol{x}_0)$ be a Gaussian mixture $h(\boldsymbol{x}_0) = \sum_{k=1}^{K} \pi_k \mathcal{N}(\boldsymbol{x}_0 \mid \mu_k, \Sigma_k)$ where $\sum_{k=1}^{K} \pi_k = 1$. The *responsibility* $r_k(\boldsymbol{x}_t)$ at point $\boldsymbol{x}_t$ of component $C_k$ is the defined as

$$r_k(\boldsymbol{x}_t) := p(C_k|\boldsymbol{x}_t) = \frac{p(\boldsymbol{x}_t|C_k)p(C_k)}{p(\boldsymbol{x}_t)}.$$

Intuitively, responsibility regions quantify which mixture component dominates (locally) at a given latent point $(\boldsymbol{x}_t, t)$. It, therefore, provides a principal way of studying the local geometry by decomposing the Fisher metric into component-wise contributions.

**Proposition 2** (Fisher metric decomposition). *Let $h(\boldsymbol{x}_0)$ be a Gaussian mixture prior. The Fisher metric at $\boldsymbol{\theta} = (\boldsymbol{x}_t, t)$ on the spacetime manifold $(\Theta, g_\alpha)$ admits the decomposition*

$$I(\boldsymbol{\theta}) = \mathbb{E}_{\boldsymbol{x}_0 \sim p(\boldsymbol{x}_0|\boldsymbol{x}_t)} \left[ \sum_{k=1}^{K} r_k(\boldsymbol{x}_t) \frac{\nabla_{\boldsymbol{\theta}} p_k(\boldsymbol{x}_0 \mid \boldsymbol{x}_t)}{p(\boldsymbol{x}_0 \mid \boldsymbol{x}_t)} \sum_{l=1}^{K} r_l(\boldsymbol{x}_t) \frac{\nabla_{\boldsymbol{\theta}} p_l(\boldsymbol{x}_0 \mid \boldsymbol{x}_t)^T}{p(\boldsymbol{x}_0 \mid \boldsymbol{x}_t)} \right] + G(\boldsymbol{\theta}),$$

*where $p_k(\boldsymbol{x}_t) = \int p(\boldsymbol{x}_t|\boldsymbol{x}_0)p_k(\boldsymbol{x}_0)d\boldsymbol{x}_0$, $p_k(\boldsymbol{x}_0) = \mathcal{N}(\boldsymbol{x}_0 \mid \mu_k, \Sigma_k)$ and $G(\boldsymbol{\theta})$ is given by*

$$
\begin{aligned}
G(\boldsymbol{\theta}) = \mathbb{E}_{\boldsymbol{x}_0 \sim p(\boldsymbol{x}_0|\boldsymbol{x}_t)} \Bigg[ &2 \sum_{k=1}^{K} \nabla_{\boldsymbol{\theta}} r_k(\boldsymbol{x}_t) \frac{p_k(\boldsymbol{x}_0 \mid \boldsymbol{x}_t)}{p(\boldsymbol{x}_0 \mid \boldsymbol{x}_t)} \sum_{l=1}^{K} r_l(\boldsymbol{x}_t) \frac{\nabla_{\boldsymbol{\theta}} p_l(\boldsymbol{x}_0 \mid \boldsymbol{x}_t)^T}{p(\boldsymbol{x}_0 \mid \boldsymbol{x}_t)} \\
&+ \sum_{k=1}^{K} \nabla_{\boldsymbol{\theta}} r_k(\boldsymbol{x}_t) \frac{p_k(\boldsymbol{x}_0 \mid \boldsymbol{x}_t)}{p(\boldsymbol{x}_0 \mid \boldsymbol{x}_t)} \sum_{l=1}^{K} \nabla_{\boldsymbol{\theta}} r_l(\boldsymbol{x}_t) \frac{p_l(\boldsymbol{x}_0 \mid \boldsymbol{x}_t)^T}{p(\boldsymbol{x}_0 \mid \boldsymbol{x}_t)} \Bigg].
\end{aligned}
\tag{3}
$$

*In particular, $G(\boldsymbol{\theta}) = 0$ whenever $\nabla_{\boldsymbol{\theta}} r_k(\boldsymbol{x}_t) = 0$ for all $k$.*

Through this decomposition (formula 3), we can see that the Fisher metric incorporates a non-conservative term $G$ arising from phase transfer, i.e. when the noise schedule induces an exchange of responsibilities (large $\nabla_{\boldsymbol{\theta}} r_k(\boldsymbol{x}_t)$). Such abrupt variations of the Fisher metric in the latent spaces of diffusion models have been studied in detail in (Lobashev et al., 2025), and relate to sudden fluctuations in image appearances (Guo et al., 2024; Liu et al., 2021), and to times of 'generative decisions' (Li and Chen, 2024).

**Proposition 3** (Negative curvature of spacetime). *For sufficiently late denoising steps, the spacetime manifold contains regions of negative sectional curvature.*

This implies that PF-ODE trajectories diverge exponentially near the clean data distribution. In contrast, for discrete data with a discrete forward process (Austin et al., 2021), the spacetime manifold would have positive constant curvature (Davis et al., 2024). This decomposition also provides a geometric perspective on phase-transition-like behavior that has been observed in diffusion models (Sclocchi et al., 2025; Ambrogioni, 2025; Raya and Ambrogioni, 2023; Yu and Huang, 2025).

The generative process can be qualitatively divided into three phases:

1. **Early denoising:** Data is highly noisy, and the responsibility regions of different components are approximately locally constant (i.e., $\nabla_{\boldsymbol{\theta}} r_k(\boldsymbol{x}_t) \approx 0$). Thus, $G(\boldsymbol{\theta}) \approx 0$. Some works suggest bypassing this initial phase using pre-trained models (Lyu et al., 2022). Our analysis thus indicates that the extent of this phase depends on the noise schedule.

2. **Intermediate denoising:** As the denoising process begins to favor specific mixture components, the responsibilities change, making the contribution of $G(\boldsymbol{\theta})$ non-negligible. This leads to a highly non-uniform geometry, which empirically manifests as ridges of sharp negative curvature (see Figure 3, and Figure 5 in Appendix 5.7). Similar phenomena were also observed in (Lobashev et al., 2025).

3. **Late denoising:** The process has effectively committed to a single mixture component, and responsibility regions become (nearly) disjoint. Thus, $\nabla_{\boldsymbol{\theta}} r_k(\boldsymbol{x}_t) \approx 0$. In this phase, the sectional curvature is negative (Proposition 3).

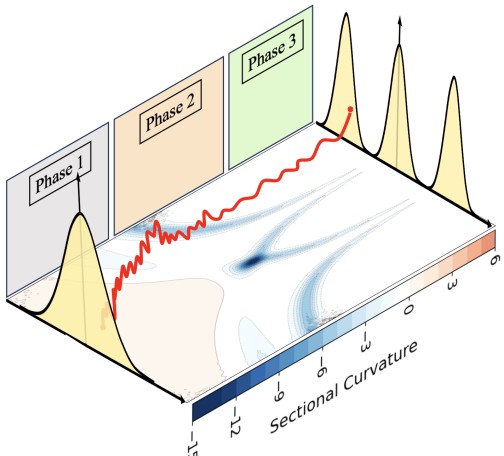

Figure 3: Phase-transition behavior (schematic).

This analysis shows how the noise schedule influences the empirical performance of diffusion models by changing both the duration and the geometry of these phases, as well as the sharpness of the transitions between them. Thus, an unsuitable choice of noise injection can lead to inefficiencies by underestimating important temporal windows. Proposition 2 also highlights that the optimal noise schedule depends on the underlying data distribution, supporting the idea that there might not be a universal optimal noise schedule across all data modalities (Dieleman et al., 2022).

## 4 CONCLUSION AND FUTURE WORK

In this paper, we have analyzed some of the intrinsic and extrinsic geometric properties of the latent space of diffusion models, emphasizing the impact of the noise schedule. This lays the foundations for future work on phase transitions and information-theoretic grounded sampling methods.

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

## 5 Appendix

### 5.1 Mathematical background

In this section, we provide the important definitions used throughout the paper.

#### Riemannian geometry

Let $\mathcal{M}$ be a smooth manifold equipped with a Riemannian metric $g$. We are interested in comparing the structure of different manifolds.

**Definition 4** (Diffeomorphic). We say that the manifolds $\mathcal{M}^{(1)}$ and $\mathcal{M}^{(2)}$ are diffeomorphic if and only if there exist a diffeomorphism $\Phi : \mathcal{M}^{(1)} \to \mathcal{M}^{(2)}$.

Note that diffeomorphisms are defined in smooth manifolds, without reference to a Riemannian metric. Once a Riemannian metric is introduced, we can define a stronger notion of equivalence.

**Definition 5** (Isometric). We say that the Riemannian manifolds $(\mathcal{M}^{(1)}, g^{(1)})$ and $(\mathcal{M}^{(2)}, g^{(2)})$ are isometric if and only if (1) there exist a diffeomorphism $\Phi : \mathcal{M}^{(1)} \to \mathcal{M}^{(2)}$, and (2) the diffeomorphism $\Phi$ preserves the metric, i.e.: $g^{(1)} = \Phi^* g^{(2)}$

We are now interested in studying properties of curves $\gamma \in \mathcal{M}$.

**Definition 6** (Length of a curve). Let $(M, g)$ be a Riemannian manifold and let $\alpha : [0, L] \to M$ be a piecewise smooth curve. The length of $\alpha$ is defined by

$$\ell(\alpha) = \int_0^L |\alpha'(t)|_g \, dt$$

where $\|\dot{\gamma}(t)\|_g = \sqrt{g_{\gamma(t)}(\dot{\gamma}(t), \dot{\gamma}(t))}$.

**Definition 7** (Energy of a curve). Let $\alpha : [0, L] \to (M, g)$ be a smooth curve. The energy of $\alpha$ is defined by

$$E(\alpha) = \int_0^L |\alpha'(t)|_g^2 \, dt.$$

Note that the length of a curve is invariant under reparametrization, whereas the energy is not, since it depends on the speed at which the curve is traversed. In particular, the following results relate these two notions:

**Lemma 2.** Let $\alpha : [0, L] \to (\mathcal{M}, g)$ be a smooth curve. Then

$$L(\alpha)^2 \leq L \, E(\alpha),$$

with equality if and only if $|\alpha'|$ is constant.

**Lemma 3.** Let $\gamma : [0, L] \to (M, g)$ be a minimizing geodesic joining $p$ and $q$. Then for all curves $\alpha : [0, L] \to M$ such that $\alpha(0) = p$ and $\alpha(L) = q$, we have

$$E(\gamma) \leq E(\alpha),$$

with equality if and only if $\alpha$ is a minimizing geodesic.

#### Information geometry

Information geometry is the study of statistical manifolds. Exponential families are a subset of these manifolds which have a particularly nice form.

**Definition 8** (Exponential family). A parametric family of probability distributions $\{p(\cdot \mid \boldsymbol{\theta})\}$ belongs to the exponential family if it can be written in the form

$$p(\boldsymbol{x}_0 \mid \boldsymbol{\theta}) = q(\boldsymbol{x}_0) \exp\left(\eta(\boldsymbol{\theta})^\top T(\boldsymbol{x}_0) - \psi(\boldsymbol{\theta})\right), \tag{4}$$

where $\eta(\boldsymbol{\theta})$ are the natural parameters, $T(\boldsymbol{x}_0)$ are the sufficient statistics, and $\psi$ is the log-partition function.

The Fisher-Rao metric

$$g^{\text{FR}}(\boldsymbol{x}_t) = \mathbb{E}_{\boldsymbol{x}_0 \sim p(\boldsymbol{x}_0 | \boldsymbol{x}_t)} \left[ \nabla_{\boldsymbol{x}_t} \log p(\boldsymbol{x}_0 \mid \boldsymbol{x}_t) \nabla_{\boldsymbol{x}_t} \log p(\boldsymbol{x}_0 \mid \boldsymbol{x}_t)^\top \right] \in \mathbb{R}^{\dim(\Theta) \times \dim(\Theta)}.$$

measures how an infinitesimal noise step $d\boldsymbol{x}_t$ changes the denoising distribution, and is related to the KL-divergence via

$$\text{KL}(p(\boldsymbol{x}_0 \mid \boldsymbol{x}_t) \,\|\, p(\boldsymbol{x}_0 \mid \boldsymbol{x}_t + d\boldsymbol{x}_t)) = \frac{1}{2} d\boldsymbol{x}_t^\top g^{\text{FR}}(\boldsymbol{x}_t) d\boldsymbol{x}_t + o\big(\|d\boldsymbol{x}_t\|^2\big).$$

For a more detail introduction to information geometry, see (Mishra et al., 2023).

**Definition 9** (Fisher speed). Let $\gamma : [0, T] \to \mathcal{M}$ be a smooth curve of probability distributions. The *Fisher speed* of the curve at time $t$ with respect to the Fisher–Rao metric is defined as

$$v_F(t) = \|\dot{\gamma}(t)\|_{g_{\gamma(t)}^{FR}} = \sqrt{g_{\gamma(t)}^{FR}(\dot{\gamma}(t), \dot{\gamma}(t))}.$$

## 5.2 PROOF OF LEMMA 1

For a given noise schedule $\{\alpha(t), \sigma(t)\}$, define

$$\iota_\alpha : \Theta \to \mathcal{M}_\alpha \subset \mathcal{P}, \qquad \iota_\alpha(\boldsymbol{x}_t, t) := p_\alpha(\boldsymbol{x}_0 \mid \boldsymbol{x}_t),$$

for $\Theta := \mathbb{R}^d \times (0, T]$. By Bayes' theorem,

$$p_\alpha(\boldsymbol{x}_0 \mid \boldsymbol{x}_t) = \frac{p_\alpha(\boldsymbol{x}_t \mid \boldsymbol{x}_0) \, h(\boldsymbol{x}_0)}{p_\alpha(\boldsymbol{x}_t)}, \qquad p_\alpha(\boldsymbol{x}_t \mid \boldsymbol{x}_0) = \mathcal{N}\big(\boldsymbol{x} \mid \alpha(t)\boldsymbol{x}_0, \, \sigma(t)^2 I\big).$$

The inverse map $\iota_\alpha^{-1} : \mathcal{M}_\alpha \to (0, T] \times \mathbb{R}^d$ is simply the projection

$$\iota_\alpha^{-1}(p(\boldsymbol{x}_0 \mid \boldsymbol{x}_t)) = (\boldsymbol{x}_t, t).$$

Both $\iota_\alpha$ and $\iota_\alpha^{-1}$ are smooth in $(\boldsymbol{x}_t, t)$, and $\iota_\alpha$ is injective since distinct pairs $(\boldsymbol{x}_t, t)$ induce distinct posterior distributions $p(\boldsymbol{x}_0 \mid \boldsymbol{x}_t)$ under a fixed diffusion kernel. One can think of $\mathcal{M}_\alpha$ as the portion of the denoising manifold explored by a diffusion process with a particular noise schedule. As $\iota_\alpha$ is a smooth homeomorphism into its image, it follows that $\mathcal{M}_\alpha$ is an embedded submanifold of $\mathcal{P}$. Its Fisher metric is just the restriction to this submanifold.

## 5.3 PROPERTIES OF THE SPACETIME MANIFOLD

In this section, we describe some properties of the spacetime manifold which might be useful to get some intuition into this construction.

**Lemma 4.** *The set of denoising distributions $p_\alpha(\boldsymbol{x}_0 \mid \boldsymbol{x}_t)$ parametrized by $\boldsymbol{\theta} = (\boldsymbol{x}_t, t)$, (i.e., $\mathcal{M}_\alpha$), forms an exponential family with the following structure*

$$p(\boldsymbol{x}_0 \mid \boldsymbol{x}_t) = h(\boldsymbol{x}_0) \exp\big(\eta(\boldsymbol{x}_t, t)^\top T(\boldsymbol{x}_0) - \psi(\boldsymbol{x}_t, t)\big), \tag{5}$$

*where $\psi$ is the log-partition function, $\eta$ the natural parameter and $T$ the sufficient statistics*

$$\begin{cases} \eta(\boldsymbol{x}_t, t) = \left( \dfrac{\alpha_t}{\sigma_t^2} \boldsymbol{x}_t, \, -\dfrac{\alpha_t^2}{2\sigma_t^2} \right), \\[2mm] T(\boldsymbol{x}_0) = \big(\boldsymbol{x}_0, \, \|\boldsymbol{x}_0\|^2\big), \\[2mm] \psi(\boldsymbol{x}_t, t) = \log p_t(\boldsymbol{x}_t) + \dfrac{D}{2} \log\big(2\pi\sigma_t^2\big) + \|\boldsymbol{x}_t\|^2. \end{cases} \tag{6}$$

*Furthermore, the expectation parameter is given by*

$$\mu(\boldsymbol{x}_t, t) = \left( \frac{1}{\alpha_t}\big(\boldsymbol{x}_t + \sigma_t^2 \nabla_{\boldsymbol{x}_t} \log p_t(\boldsymbol{x}_t)\big), \, \frac{\sigma_t^2}{\alpha_t} \text{div}_{\boldsymbol{x}_t} \mathbb{E}[\boldsymbol{x}_0 \mid \boldsymbol{x}_t] + \Big\|\mathbb{E}[\boldsymbol{x}_0 \mid \boldsymbol{x}_t]\Big\|^2 \right).$$

*Proof.* The proof can be found in (Karczewski et al., 2026). $\qquad\square$

This property simplifies the geometry of the manifold and yields a practical method for computing geodesics between any two samples through the spacetime. In addition, this form will be important for the proof of Lemma 1 in Appendix 5.4.

**Remark 1.** Note that $\mathcal{M}$ is not a *natural* exponential family in the sense that we cannot express $\eta(\boldsymbol{x}_t, t)$ linearly in terms of $(\boldsymbol{x}_t, t)$. Natural exponential families are particularly convenient because the Fisher metric tensor can be derived directly from $\psi$, and the Riemannian curvature tensor can be computed through determinant computations involving partial derivatives of the metric.

**Remark 2.** The family $\mathcal{M}$ forms a manifold with boundary. At $t = T$, we have $p(\boldsymbol{x}_0 \mid \boldsymbol{x}_T) \approx h(\boldsymbol{x}_0)$, meaning the conditional distribution is approximately independent of $\boldsymbol{x}_T$. Consequently, $\nabla_{\boldsymbol{x}_T} \log p(\boldsymbol{x}_0 \mid \boldsymbol{x}_T) \approx 0$, and the Fisher information metric becomes degenerate at this boundary.

On the other hand, as $t \to 0$, the derivative $\frac{\partial \eta(\boldsymbol{\theta})}{\partial \boldsymbol{\theta}}$ diverges as $\sigma(t) \to 0$, causing the Fisher metric to blow up. Thus, the boundary at $t = 0$ lies at an *infinite* distance.

### 5.4 PROOF OF PROPOSITION 1

To prove Proposition 1, we will first introduce the following results.

**Lemma 5** (Invariance of the Fisher–Rao metric under reparametrization). *Let $\Phi : \Theta \to \Theta'$ be a smooth reparametrization and set $\boldsymbol{\theta}' = \Phi(\boldsymbol{\theta})$ such that*

$$p(\boldsymbol{x} \mid \boldsymbol{\theta}) = p(\boldsymbol{x} \mid \boldsymbol{\theta}') = p(\boldsymbol{x} \mid \Phi(\boldsymbol{\theta})), \qquad \text{for all } \boldsymbol{\theta} \text{ and all } \boldsymbol{x}.$$

*Then $\Phi$ is an isometry between the statistical manifolds $(\Theta, g)$ and $(\Theta', g')$, i.e. for all $\boldsymbol{\theta} \in \Theta$ and all $u, v \in T_{\boldsymbol{\theta}}\Theta$,*

$$g_{\boldsymbol{\theta}}(u, v) = g'_{\boldsymbol{\theta}'}\big(d\Phi_{\boldsymbol{\theta}} u, \ d\Phi_{\boldsymbol{\theta}} v\big).$$

*Equivalently, the Fisher information matrices satisfy*

$$I_{\boldsymbol{\theta}} = J_{\Phi}(\boldsymbol{\theta})^{\top} I_{\boldsymbol{\theta}'}(\Phi(\boldsymbol{\theta})) \, J_{\Phi}(\boldsymbol{\theta}),$$

*where $J_{\Phi}(\boldsymbol{\theta})$ denotes the Jacobian of $\Phi$ at $\boldsymbol{\theta}$.*

*Proof.* Fix $\boldsymbol{\theta} \in \Theta$ and set $\boldsymbol{\theta}' = \Phi(\boldsymbol{\theta})$. By assumption, for all $\boldsymbol{x}$,

$$\log p(\boldsymbol{x} \mid \boldsymbol{\theta}) = \log p(\boldsymbol{x} \mid \boldsymbol{\theta}') = \log p(\boldsymbol{x} \mid \Phi(\boldsymbol{\theta})).$$

Differentiating with respect to $\boldsymbol{\theta}$ and applying the chain rule yields

$$\nabla_{\boldsymbol{\theta}} \log p(\boldsymbol{x} \mid \boldsymbol{\theta}) = J_{\Phi}(\boldsymbol{\theta})^{\top} \nabla_{\boldsymbol{\theta}'} \log p(\boldsymbol{x} \mid \boldsymbol{\theta}').$$

The Fisher information at $\boldsymbol{\theta}$ is therefore

$$
\begin{aligned}
I_{\boldsymbol{\theta}} &= \mathbb{E}_{\boldsymbol{x} \sim p(\cdot|\boldsymbol{\theta})}\Big[\nabla_{\boldsymbol{\theta}} \log p(\boldsymbol{x} \mid \boldsymbol{\theta})\nabla_{\boldsymbol{\theta}} \log p(\boldsymbol{x} \mid \boldsymbol{\theta})^{\top}\Big] \\
&= \mathbb{E}_{\boldsymbol{x} \sim p(\cdot|\boldsymbol{\theta})}\Big[J_{\Phi}(\boldsymbol{\theta})^{\top} \nabla_{\boldsymbol{\theta}'} \log p(\boldsymbol{x} \mid \boldsymbol{\theta}')\nabla_{\boldsymbol{\theta}'} \log p(\boldsymbol{x} \mid \boldsymbol{\theta}')^{\top} J_{\Phi}(\boldsymbol{\theta})\Big] \\
&= J_{\Phi}(\boldsymbol{\theta})^{\top} \mathbb{E}_{\boldsymbol{x} \sim p(\cdot|\boldsymbol{\theta}')}\Big[\nabla_{\boldsymbol{\theta}'} \log p(\boldsymbol{x} \mid \boldsymbol{\theta}')\nabla_{\boldsymbol{\theta}'} \log p(\boldsymbol{x} \mid \boldsymbol{\theta}')^{\top}\Big] J_{\Phi}(\boldsymbol{\theta}).
\end{aligned}
$$

Since $p(\boldsymbol{x} \mid \boldsymbol{\theta}) = p(\boldsymbol{x} \mid \boldsymbol{\theta}')$, the expectation equals $I_{\boldsymbol{\theta}'}(\boldsymbol{\theta}') = I_{\boldsymbol{\theta}'}(\Phi(\boldsymbol{\theta}))$, giving

$$I_{\boldsymbol{\theta}} = J_{\Phi}(\boldsymbol{\theta})^{\top} I_{\boldsymbol{\theta}'}(\Phi(\boldsymbol{\theta})) \, J_{\Phi}(\boldsymbol{\theta}).$$

Finally, for any $u, v \in T_{\boldsymbol{\theta}}\Theta$,

$$
\begin{aligned}
g'_{\boldsymbol{\theta}'}(d\Phi_{\boldsymbol{\theta}} u, \ d\Phi_{\boldsymbol{\theta}} v) &= (J_{\Phi}(\boldsymbol{\theta})u)^{\top} I_{\boldsymbol{\theta}'}(\Phi(\boldsymbol{\theta}))(J_{\Phi}(\boldsymbol{\theta})v) \\
&= u^{\top}\big(J_{\Phi}(\boldsymbol{\theta})^{\top} I_{\boldsymbol{\theta}'}(\Phi(\boldsymbol{\theta}))J_{\Phi}(\boldsymbol{\theta})\big)v \\
&= u^{\top} I_{\boldsymbol{\theta}} v = g_{\boldsymbol{\theta}}(u, v),
\end{aligned}
$$

which concludes the proof. $\qquad\square$

**Lemma 6** (Identifiability of natural parameters). *Let $\{p(\boldsymbol{x}_0 \mid \boldsymbol{\theta}) : \boldsymbol{\theta} \in \Theta\}$ be an exponential family of the form*

$$p(\boldsymbol{x}_0 \mid \boldsymbol{\theta}) = h(\boldsymbol{x}_0) \exp\big(\eta(\boldsymbol{\theta})^\top T(\boldsymbol{x}_0) - \psi(\boldsymbol{\theta})\big),$$

*where $\eta(\boldsymbol{\theta})$ denotes the natural parameter. For a fixed $h(\boldsymbol{x}_0)$ and $T(\boldsymbol{x}_0)$, if $\eta(\boldsymbol{\theta}_1) = \eta(\boldsymbol{\theta}_2)$, then*

$$p(\boldsymbol{x}_0 \mid \boldsymbol{\theta}_1) = p(\boldsymbol{x}_0 \mid \boldsymbol{\theta}_2) \quad \text{for all } \boldsymbol{x}_0.$$

*In particular, the natural parameter $\eta(\boldsymbol{\theta})$ uniquely determines the probability distribution.*

*Proof.* Fix $\boldsymbol{\theta} \in \Theta$. Since $p(\cdot \mid \boldsymbol{\theta})$ is a probability density function, it must satisfy the

$$\int p(\boldsymbol{x}_0 \mid \boldsymbol{\theta}) \, d\boldsymbol{x}_0 = 1.$$

Substituting the exponential family form yields

$$1 = \int h(\boldsymbol{x}_0) \exp\big(\eta(\boldsymbol{\theta})^\top T(\boldsymbol{x}_0) - \psi(\boldsymbol{\theta})\big) \, d\boldsymbol{x}_0 = e^{-\psi(\boldsymbol{\theta})} \int h(\boldsymbol{x}_0) \exp\big(\eta(\boldsymbol{\theta})^\top T(\boldsymbol{x}_0)\big) \, d\boldsymbol{x}_0.$$

Solving for $\psi(\boldsymbol{\theta})$ gives

$$\psi(\boldsymbol{\theta}) = \log \int h(\boldsymbol{x}_0) \exp\big(\eta(\boldsymbol{\theta})^\top T(\boldsymbol{x}_0)\big) \, d\boldsymbol{x}_0.$$

By assumption, $h(\boldsymbol{x}_0)$ and $T(\boldsymbol{x}_0)$ are fixed, so if $\eta(\boldsymbol{\theta}_1) = \eta(\boldsymbol{\theta}_2)$ then if $\psi(\boldsymbol{\theta}_1) = \psi(\boldsymbol{\theta}_2)$ which implies that $p(\boldsymbol{x}_0 \mid \boldsymbol{\theta}_1) = p(\boldsymbol{x}_0 \mid \boldsymbol{\theta}_2)$ for all $\boldsymbol{x}_0$. □

**Lemma 7.** *Consider the VP gaussian diffusion process $x(t) = \alpha(t)\boldsymbol{x}_0 + \sigma(t)\boldsymbol{x}_T$. Assume $\alpha_t$ is monotonic. Then a change of noise schedule is mathematically equivalent to a change of time parametrization.*

*Proof.* Let $\alpha_1(t)$ and $\alpha_2(t)$ define different noise schedules. Take the time parametrization $\phi(t) = \alpha_1^{-1} \circ \alpha_2(t)$. Then, $\alpha_1(\phi(t)) = \alpha_2(t)$. Note that the inverse is well-defined as we assumed $\alpha_1$ is monotone. □

Here starts the proof of Proposition 1:

*Proof.* First note that in a diffusion process, $\boldsymbol{x}$ is disentangled from $t$. So the correct definition of the coordinate chart is $(\boldsymbol{x}, t)$ instead of $(\boldsymbol{x}_t, t)$. We have used the latter one in the main text to stick with the standard notion of diffusion models. Nevertheless, this distinction will be important for this proof.

**(ii)** $\Rightarrow$ **(i)** Assume that there exists a proper time change $\phi : [0, T] \to [0, T]$ relating the two noise schedules. Then

$$\begin{cases} \alpha_1(\phi(t)) = \alpha_2(t) \\ \sigma_1(\phi(t)) = \sigma_2(t) \end{cases} \tag{7}$$

for all $t$, and is boundary preserving. For each schedule $i \in \{1, 2\}$, we consider the respective distribution in exponential form

$$p_t^{(i)}(\boldsymbol{x}_0 \mid \boldsymbol{x}) \propto h(\boldsymbol{x}_0) \exp\big(\eta^{(i)}(\boldsymbol{x}, t) T(\boldsymbol{x}_0)\big)$$

with natural parameters $\eta(\boldsymbol{x}, t) = \left(\dfrac{\alpha_t}{\sigma_t^2}\boldsymbol{x}, -\dfrac{\alpha_t^2}{2\sigma_t^2}\right)$, as defined in Lemma 4. Finding a relationship between the manifolds reduces to finding a relationship between the posterior distributions and hence, by Lemma 6, the natural parameters. So we want to build a smooth, invertible change of variables $\Phi : \mathcal{M}_{\alpha_1} \to \mathcal{M}_{\alpha_2}$ such that $\eta^{(1)} \circ \Psi = \eta^{(2)}$. Consider the map

$$\Psi(\boldsymbol{x}, t) := (\boldsymbol{x}, \phi(t)).$$

In coordinates $(\boldsymbol{x}, t) \in \mathbb{R}^{D+1}$, its Jacobian has block form

$$J_\Psi(\boldsymbol{x}, t) = \begin{pmatrix} \dfrac{\partial \boldsymbol{x}}{\partial \boldsymbol{x}} & \dfrac{\partial \boldsymbol{x}}{\partial t} \\ \dfrac{\partial \phi(t)}{\partial \boldsymbol{x}} & \dfrac{\partial \phi(t)}{\partial t} \end{pmatrix} = \begin{pmatrix} I_D & 0_{D \times 1} \\ 0_{1 \times D} & \phi'(t) \end{pmatrix},$$

with determinant $\det J_\Phi(\boldsymbol{x}, t) = \phi'(t)$. By assumption, $\phi'(t) \neq 0$ for all $t$, hence $J_\Phi$ is invertible everywhere. Therefore, $\Phi$ is a local $C^2$ diffeomorphism. Global invertibility follows from the strict monotonicity of $\phi$, with inverse map

$$\Psi^{-1}(\boldsymbol{x}, \phi(t)) := \big(\boldsymbol{x}, \ \phi^{-1}(\phi(t)) = (\boldsymbol{x}, t).$$

Then, by equation 7, $\eta^{(1)}(\Psi(\boldsymbol{x}, t)) = \eta^{(2)}(x, t)$, so that $p_t^{(1)}(\boldsymbol{x}_0 \mid \boldsymbol{x}) = p_{\phi(t)}^{(2)}(\boldsymbol{x}_0 \mid \boldsymbol{x})$. Hence, $\Psi$ is a $C^2$ diffeomorphism between $\mathcal{M}_{\alpha_1}$ and $\mathcal{M}_{\alpha_2}$. To prove isometry, note that by Lemma 7, every change of noise is equivalent to a time reparametrization. By Lemma 5, any coordinate reparameterization induces an isometry.

**(i)** $\Rightarrow$ **(ii)** Conversely, assume that the Riemannian manifolds $(\mathcal{M}_{\alpha_1}, g_1^{FR})$ and $(\mathcal{M}_{\alpha_2}, g_2^{FR})$ are isometric. Then there exists a smooth, invertible, boundary-preserving map $\Phi : \mathcal{M}_{\alpha_1} \to \mathcal{M}_{\alpha_2}$ such that $\eta^{(1)} \circ \Phi = \eta^{(2)}$, by the identifiability of the natural parameters (Lemma 6). Since a change in noise schedule induces a reparametrization of time, there exists a smooth and invertible function $\phi : [0, T] \to [0, T]$ such that $\Phi(\boldsymbol{x}, t) = (\boldsymbol{x}, \phi(t))$. Moreover, since $\Phi$ is boundary preserving, $\phi(0) = 0$ and $\phi(T) = T$. Thus, $\phi$ is a proper time change. $\qquad \square$

Establishing the geometric equivalence between $\mathcal{M}_{\alpha_1}$ and $\mathcal{M}_{\alpha_2}$ in Proposition 1 ensures that the pullback metrics associated with different noise schedules are defined on the same underlying family of denoising distributions, rather than on genuinely distinct statistical models. Consequently, any difference in the geometry of the *spacetime* manifold arises through the distortion induced by different noise schedules embeddings, which change the pullback metric in a *non-trivial* way.

## 5.5 Equivalence of SDEs under proper time changes

This is a similar argument to the one presented in (Stancevic et al., 2025). Consider the forward diffusion process

$$dX_t = f(X_t, t)\, dt + g(t)\, dW_t,$$

with $f(t) = \frac{d}{dt} \log \alpha(t)$ and $g(t) = -\sigma^2(t) \frac{d\lambda_t}{dt}$, where $\lambda(t) = \log \mathrm{SNR}(t)$. Consider two noise schedules $\{\alpha_i(t), \sigma_i(t)\}_{t \in (0, T]}$ for $i \in \{1, 2\}$ related by a proper time change $\phi : t \mapsto s$. Their respective Reverse SDE are:

$$\begin{aligned} &1. \quad dX_t = f_1(X_t, t)\, dt + g_1(t)\, dW_t, \\ &2. \quad dY_s = f_2(Y_s, s)\, ds + g_2(s)\, dW_s. \end{aligned} \tag{8}$$

If two noise schedules are related by a proper time change, then

$$\alpha_1(\phi(t)) = \alpha_2(s) \quad \text{and} \quad \sigma_1(\phi(t))) = \alpha_2(s) \quad \forall t \in [0, T].$$

Thus,

$$f_2(s) = \frac{d}{ds} \log \alpha_2(s) = \frac{d}{dt} \log \alpha_1(\phi(t)) = f_1(\phi(t))\phi'(t)$$

and

$$g_2(s) = -\sigma_2^2(s) \frac{d}{ds} \frac{\alpha_2^2(s)}{\sigma_2^2(s)} = -\sigma_1^2(\phi(t)) \frac{d}{dt} \frac{\alpha_1^2(\phi(t))}{\sigma_1^2(\phi(t))} = \sqrt{\phi'(t)} g_1(\phi(t)).$$

Equation 9 can then be rewritten as

$$\begin{aligned} &1. \quad dX_t = f_1(X_t, t)\, dt + g_1(t)\, dW_t, \\ &2. \quad dY_t = f_1(Y_t, \phi(t))\phi'(t)\, dt + \sqrt{\phi'(t)} g_1(\phi(t))\, dW_t. \end{aligned} \tag{9}$$

It can now be shown using classical results from SDE that if $X_t$ solves (1), then $Y_t := X_{\phi(t)}$ solves (2). Thus, the forward dynamical system of two SDE related by proper time parametrization is not fundamentally different. In particular, the forward kernels remain the same, in the sense that

$$p_t(\boldsymbol{x} \mid \boldsymbol{x}_0) = p_{\phi(t)}(\boldsymbol{x} \mid \boldsymbol{x}_0),$$

which, by Bayes's rule, implies that

$$p_t(\boldsymbol{x}_0 \mid \boldsymbol{x}) = p_{\phi(t)}(\boldsymbol{x} \mid \boldsymbol{x}_0).$$

### 5.6 PROOF OF PROPOSITION 2

**Lemma 8.** *Consider the Gaussian mixture prior $h(\boldsymbol{x}_0)$ and the forward process $p(\boldsymbol{x}_t \mid \boldsymbol{x}_0) = \mathcal{N}\left(\boldsymbol{x}_t \mid \alpha_t \boldsymbol{x}_0, \ \sigma_t^2 I\right)$. Then, the denoising distribution $p(\boldsymbol{x}_0 \mid \boldsymbol{x}_t)$ is also a Gaussian mixture.*

*Proof.* The prior $h$ is defined as a mixture of Gaussians: $h(\boldsymbol{x}_0) = \sum_{k=1}^{K} \pi_k \mathcal{N}(\boldsymbol{x}_0 \mid \mu_k, \Sigma_k)$ with $\sum_{k=1}^{K} \pi_k = 1$, and for simplicity $p_k(\boldsymbol{x}_0) = \mathcal{N}(\boldsymbol{x}_0 \mid \mu_k, \Sigma_k)$.

Using Bayes, note that we have, for a fixed $k$: $p(\boldsymbol{x}_t|\boldsymbol{x}_0)p_k(\boldsymbol{x}_0) = p_k(\boldsymbol{x}_t)p_k(\boldsymbol{x}_0|\boldsymbol{x}_t)$, with $p_k(\boldsymbol{x}_t)$ and $p_k(\boldsymbol{x}_0|\boldsymbol{x}_t)$ to be defined. Borrowing the equations from (Bishop and Nasrabadi, 2006, eqs 2.113-2.117), we know that if

$$p_k(\boldsymbol{x}_0) = \mathcal{N}(\boldsymbol{x}_0|\mu_k, \Sigma_k),$$
$$p(\boldsymbol{x}_t|\boldsymbol{x}_0) = \mathcal{N}(\boldsymbol{x}_t|\alpha_t \boldsymbol{x}_0, \sigma_t^2 I)$$

then

$$p_k(\boldsymbol{x}_t) = \mathcal{N}(\boldsymbol{x}_t|\alpha_t \mu_k, \alpha_t^2 \Sigma_k + \sigma_t^2 I),$$
$$p_k(\boldsymbol{x}_0|\boldsymbol{x}_t) = \mathcal{N}(\boldsymbol{x}_0|\Lambda\{\alpha_t \sigma_t^{-2}\boldsymbol{x}_t I + \Sigma_k^{-1}\mu_k\}, \Lambda)$$

with $\Lambda := \left(\Sigma_k^{-1} + \alpha_t^2 \sigma_t^{-2}\right)^{-1}$. In addition,

$$p(\boldsymbol{x}_t) = \int p(\boldsymbol{x}_t \mid \boldsymbol{x}_0) \, h(\boldsymbol{x}_0) \, d\boldsymbol{x}_0 = \int p(\boldsymbol{x}_t \mid \boldsymbol{x}_0) \left( \sum_j \pi_j \, p_j(\boldsymbol{x}_0) \right) d\boldsymbol{x}_0$$

$$= \sum_j \pi_j \int p(\boldsymbol{x}_t \mid \boldsymbol{x}_0) \, p_j(\boldsymbol{x}_0) \, d\boldsymbol{x}_0 = \sum_j \pi_j \, p_j(\boldsymbol{x}_t).$$

and similarly

$$p(\boldsymbol{x}_t|\boldsymbol{x}_0)h(\boldsymbol{x}_0) = \sum_k \pi_k \, p(\boldsymbol{x}_t|\boldsymbol{x}_0)p_k(\boldsymbol{x}_0) = \sum_k \pi_k \, p_k(\boldsymbol{x}_0|\boldsymbol{x}_t)p_k(\boldsymbol{x}_t).$$

Thus,

$$p(\boldsymbol{x}_0|\boldsymbol{x}_t) = \frac{p(\boldsymbol{x}_t|\boldsymbol{x}_0)h(\boldsymbol{x}_0)}{p(\boldsymbol{x}_t)} = \frac{\sum_k \pi_k \, p_k(\boldsymbol{x}_0|\boldsymbol{x}_t)p_k(\boldsymbol{x}_t)}{\sum_j \pi_j \, p_j(\boldsymbol{x}_t)} = \sum_k \left( \frac{\pi_k \, p_k(\boldsymbol{x}_t)}{\sum_j \pi_j \, p_j(\boldsymbol{x}_t)} \right) p_k(\boldsymbol{x}_0|\boldsymbol{x}_t).$$

So $p(\boldsymbol{x}_0|\boldsymbol{x}_t)$ can be expressed as a mixture of Gaussians

$$p(\boldsymbol{x}_0|\boldsymbol{x}_t) = \sum_k \gamma_k(\boldsymbol{x}_t)p_k(\boldsymbol{x}_0|\boldsymbol{x}_t)$$

with $p_k(\boldsymbol{x}_0|\boldsymbol{x}_t) = \mathcal{N}(\boldsymbol{x}_0|\Lambda\{\frac{\alpha_t}{\sigma_t^2}\boldsymbol{x}_t I + \Sigma_k^{-1}\mu_k\}, \Lambda)$, $\Lambda = \left(\Sigma_k^{-1} + \frac{\alpha_t^2}{\sigma_t^2}\right)^{-1}$ and $\sum_k \gamma_k(\boldsymbol{x}_t) = 1$. $\square$

**Lemma 9.** *The denoising distributions $p(\boldsymbol{x}_0|\boldsymbol{x}_t)$ can be expressed in terms of the responsibility regions.*

*Proof.* By Bayes' Theorem,

$$p(\boldsymbol{x}_0|\boldsymbol{x}_t) = \frac{p(\boldsymbol{x}_t|\boldsymbol{x}_0)h(\boldsymbol{x}_0)}{p(\boldsymbol{x}_t)} = \sum_{k=1}^{K} \frac{\pi_k p_k(\boldsymbol{x}_t)}{p(\boldsymbol{x}_t)} \frac{p(\boldsymbol{x}_t|\boldsymbol{x}_0)p_k(\boldsymbol{x}_0)}{p_k(\boldsymbol{x}_t)} = \sum_{k=1}^{K} r_k(\boldsymbol{x}_t)p_k(\boldsymbol{x}_0|\boldsymbol{x}_t),$$

where $p_k(\boldsymbol{x}_t) = \int p(\boldsymbol{x}_t|\boldsymbol{x}_0)p_k(\boldsymbol{x}_0)d\boldsymbol{x}_0$ and $p_k(\boldsymbol{x}_0) = \mathcal{N}(\boldsymbol{x}_0 \mid \mu_k, \Sigma_k)$. $\square$

Here starts the proof of Proposition 2.

*Proof.* From Definition 2, we have that

$$I(\boldsymbol{\theta}) = \mathbb{E}_{\boldsymbol{x}_0 \sim p(\boldsymbol{x}_0 | \boldsymbol{x}_t)} \big[ \nabla_{\boldsymbol{\theta}} \log p(\boldsymbol{x}_0 \mid \boldsymbol{x}_t) \, \nabla_{\boldsymbol{\theta}} \log p(\boldsymbol{x}_0 \mid \boldsymbol{x}_t)^{\top} \big] \in \mathbb{R}^{\dim(\Theta) \times \dim(\Theta)}.$$

where $\boldsymbol{\theta} = (\boldsymbol{x}_t, t)$. Thus we get

$$I(\boldsymbol{\theta}) := \mathbb{E}_{\boldsymbol{x}_0 \sim p(\boldsymbol{x}_0 | \boldsymbol{x}_t)} \left[ \nabla_{\boldsymbol{\theta}} \log p(\boldsymbol{x}_0 \mid \boldsymbol{x}_t) \, \nabla_{\boldsymbol{\theta}} \log p(\boldsymbol{x}_0 \mid \boldsymbol{x}_t)^{\top} \right]$$

$$= \mathbb{E}_{\boldsymbol{x}_0 \sim p(\boldsymbol{x}_0 | \boldsymbol{x}_t)} \left[ \frac{\nabla_{\boldsymbol{\theta}} p(\boldsymbol{x}_0 \mid \boldsymbol{x}_t)}{p(\boldsymbol{x}_0 \mid \boldsymbol{x}_t)} \, \frac{\nabla_{\boldsymbol{\theta}} p(\boldsymbol{x}_0 \mid \boldsymbol{x}_t)^{\top}}{p(\boldsymbol{x}_0 \mid \boldsymbol{x}_t)} \right]$$

$$= \mathbb{E}_{\boldsymbol{x}_0 \sim p(\boldsymbol{x}_0 | \boldsymbol{x}_t)} \left[ \frac{\nabla_{\boldsymbol{\theta}} \left( \sum_{k=1}^{K} r_k(\boldsymbol{x}_t) \, p_k(\boldsymbol{x}_0 \mid \boldsymbol{x}_t) \right)}{p(\boldsymbol{x}_0 \mid \boldsymbol{x}_t)} \, \frac{\nabla_{\boldsymbol{\theta}} \left( \sum_{l=1}^{K} r_l(\boldsymbol{x}_t) \, p_l(\boldsymbol{x}_0 \mid \boldsymbol{x}_t) \right)^{\top}}{p(\boldsymbol{x}_0 \mid \boldsymbol{x}_t)} \right]$$

$$= \mathbb{E}_{\boldsymbol{x}_0 \sim p(\boldsymbol{x}_0 | \boldsymbol{x}_t)} \left[ \frac{1}{p(\boldsymbol{x}_0 \mid \boldsymbol{x}_t)^2} \left( \sum_{k=1}^{K} \big[ (\nabla_{\boldsymbol{\theta}} r_k(\boldsymbol{x}_t)) \, p_k(\boldsymbol{x}_0 \mid \boldsymbol{x}_t) + r_k(\boldsymbol{x}_t) \, \nabla_{\boldsymbol{\theta}} p_k(\boldsymbol{x}_0 \mid \boldsymbol{x}_t) \big] \right) \right.$$
$$\left. \cdot \left( \sum_{l=1}^{K} \big[ (\nabla_{\boldsymbol{\theta}} r_l(\boldsymbol{x}_t)) \, p_l(\boldsymbol{x}_0 \mid \boldsymbol{x}_t) + r_l(\boldsymbol{x}_t) \, \nabla_{\boldsymbol{\theta}} p_l(\boldsymbol{x}_0 \mid \boldsymbol{x}_t) \big]^{\top} \right) \right]$$

$$= \mathbb{E}_{\boldsymbol{x}_0 \sim p(\boldsymbol{x}_0 | \boldsymbol{x}_t)} \left[ \frac{\sum_{k=1}^{K} r_k(\boldsymbol{x}_t) \, \nabla_{\boldsymbol{\theta}} p_k(\boldsymbol{x}_0 \mid \boldsymbol{x}_t)}{p(\boldsymbol{x}_0 \mid \boldsymbol{x}_t)} \, \frac{\sum_{l=1}^{K} r_l(\boldsymbol{x}_t) \, \nabla_{\boldsymbol{\theta}} p_l(\boldsymbol{x}_0 \mid \boldsymbol{x}_t)^{\top}}{p(\boldsymbol{x}_0 \mid \boldsymbol{x}_t)} \right]$$

$$+ 2 \, \mathbb{E}_{\boldsymbol{x}_0 \sim p(\boldsymbol{x}_0 | \boldsymbol{x}_t)} \left[ \frac{\sum_{k=1}^{K} (\nabla_{\boldsymbol{\theta}} r_k(\boldsymbol{x}_t)) \, p_k(\boldsymbol{x}_0 \mid \boldsymbol{x}_t)}{p(\boldsymbol{x}_0 \mid \boldsymbol{x}_t)} \, \frac{\sum_{l=1}^{K} r_l(\boldsymbol{x}_t) \, \nabla_{\boldsymbol{\theta}} p_l(\boldsymbol{x}_0 \mid \boldsymbol{x}_t)^{\top}}{p(\boldsymbol{x}_0 \mid \boldsymbol{x}_t)} \right]$$

$$+ \mathbb{E}_{\boldsymbol{x}_0 \sim p(\boldsymbol{x}_0 | \boldsymbol{x}_t)} \left[ \frac{\sum_{k=1}^{K} (\nabla_{\boldsymbol{\theta}} r_k(\boldsymbol{x}_t)) \, p_k(\boldsymbol{x}_0 \mid \boldsymbol{x}_t)}{p(\boldsymbol{x}_0 \mid \boldsymbol{x}_t)} \, \frac{\sum_{l=1}^{K} (\nabla_{\boldsymbol{\theta}} r_l(\boldsymbol{x}_t)) \, p_l(\boldsymbol{x}_0 \mid \boldsymbol{x}_t)^{\top}}{p(\boldsymbol{x}_0 \mid \boldsymbol{x}_t)} \right],$$

where in the second equality we have used the decomposition of $p(\boldsymbol{x}_0 | \boldsymbol{x}_t)$ in responsibility regions, and then the product rule. Note that whenever $\nabla_{\boldsymbol{\theta}} r_k(\boldsymbol{x}_t) = 0$ for all $k$, then the Fisher metric simplifies to

$$I_{\boldsymbol{\theta}} = \mathbb{E}_{\boldsymbol{x}_0 \sim p(\boldsymbol{x}_0 | \boldsymbol{x}_t)} \left[ \frac{\sum_{k=1}^{K} r_k(\boldsymbol{x}_t) \nabla_{\boldsymbol{\theta}} p_k(\boldsymbol{x}_0 \mid \boldsymbol{x}_t)}{p(\boldsymbol{x}_0 \mid \boldsymbol{x}_t)} \, \frac{\nabla_{\boldsymbol{\theta}} \sum_{l=1}^{K} r_l(\boldsymbol{x}_t) \nabla_{\boldsymbol{\theta}} p_l(\boldsymbol{x}_0 \mid \boldsymbol{x}_t)^{\top}}{p(\boldsymbol{x}_0 \mid \boldsymbol{x}_t)} \right]. \qquad (10)$$

$\square$

**Corollary 1.** *Equation 10 can in turn be decomposed as*

$$\sum_k r_k^2(\boldsymbol{x}_t) I(\boldsymbol{\theta})_k + \mathbb{E}_{\boldsymbol{x}_0 \sim p(\boldsymbol{x}_0 | \boldsymbol{x}_t)} \sum_{k \neq l} r_k(\boldsymbol{x}_t) r_l(\boldsymbol{x}_t) \nabla_{\boldsymbol{\theta}} \log p_k(\boldsymbol{x}_0 \mid \boldsymbol{x}_t) \nabla_{\boldsymbol{\theta}} \log p_l(\boldsymbol{x}_0 \mid \boldsymbol{x}_t)^{T}, \quad (11)$$

*where $I(\boldsymbol{\theta})_k$ is the Fisher metric of the k-th component, i.e.,*

$$I(\boldsymbol{\theta})_k = \mathbb{E}_{\boldsymbol{x}_0 \sim p_k(x0 | \boldsymbol{x}_t)} \left[ \nabla_{\boldsymbol{\theta}} \log p_k(\boldsymbol{x}_0 | \boldsymbol{x}_t) \, \nabla_{\boldsymbol{\theta}} \log p_k(\boldsymbol{x}_0 | \boldsymbol{x}_t)^{\top} \right].$$

Intuitively, the second term of 11 takes into account the interaction between the intersecting support of different components of the Gaussian mixture $p(\boldsymbol{x}_0 | \boldsymbol{x}_t)$. In particular, whenever the support of the components $k$ and $l$ is (nearly) disjoint, then that term cancels out.

**Lemma 10.** *Let $\mathcal{N}(\mu, \Sigma)$ be a d-dimensional Gaussian distribution equipped with the Fisher information metric. If $\Sigma = \sigma^2 I_d$ (isotropic) with $d \geq 1$, then the resulting Fisher manifold has constant negative sectional curvature, and is (up to a constant rescaling of the metric) isometric to hyperbolic space $\mathbb{H}^{d+1}$.*

*Proof.* The sectional curvature for univariate Gaussians has been derived by (Skovgaard, 1984) and (Amari and Nagaoka, 2000). □

The the proof of Proposition 3 then follows from the fact that Gaussian distributions equipped with the Fisher metric induce a hyperbolic geometry (Lemma 10). By decomposing the metric into responsibility regions, we are able to detect these regions of non-positive curvature whenever the responsibility of a Gaussian component is dominating.

## 5.7 EXPERIMENTAL DETAILS

For Figure 2, we consider a Gaussian mixture prior $h_0 = \sum_{i=1}^{3} \pi_i \mathcal{N}(\mu_i, \sigma^2)$ with $\mu_1 = -1$, $\mu_1 = 0$, $\mu_3 = 1$, $\pi_1 = 1/3$, $\pi_2 = 1/3$, $\pi_3 = 1/3$ and $\sigma = 0.001$. In this example, the data is 1D, while the spacetime is 2D. Below, we describe each noise schedule in detail:

**Cosine.** Following (Nichol and Dhariwal, 2021), we define

$$f(t) = \cos^2 \left( \frac{(t+s)\pi}{2(1+s)} \right), \quad f_0 = \cos^2 \left( \frac{s\pi}{2(1+s)} \right),$$

and set

$$\alpha^2(t) = \frac{f(t)}{f_0}, \qquad \sigma^2(t) = 1 - \alpha^2(t),$$

where $s = 0.008$ is a small offset.

**Truncated linear.** A piecewise linear schedule

$$\sigma^2(t) = \tau(t), \qquad \alpha(t) = 1 - \tau(t)$$

with a nonlinear time change $\tau(t)$

$$\tau(t) = \begin{cases} 2\sqrt{\kappa d}\, t, & t \in [0, 1/2], \\ \sqrt{\kappa d} + (1 - \sqrt{\kappa d})(2t - 1), & t \in [1/2, 1]. \end{cases}$$

We set $\kappa = 0.5$.

**Sigmoid.** We apply the sigmoid function to a linear-in-log-SNR noise schedule. That is,

$$\lambda_t := \log \mathrm{SNR}(t) = \lambda_{\max} + (\lambda_{\min} - \lambda_{\max})\, t$$

and

$$\alpha_t^2 = \sigma(\lambda_t), \qquad \sigma_t^2 = \sigma(-\lambda_t),$$

where $\sigma(\cdot)$ is the sigmoid function. We set $\lambda_{\min} = -10$, $\lambda_{\max} = 10$.

**Linear.** We define the linear schedule

$$\sigma^2(t) = t, \qquad \alpha^2(t) = 1 - \sigma^2(t).$$

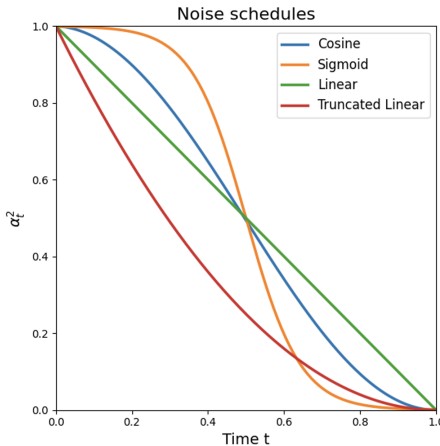

Figure 4: Noise schedules used in the simulations

Since $h_0$ is a Gaussian mixture, all marginals $p_t$ are also Gaussian mixtures. So there is no need to train a diffusion model, as the score function $\nabla_x \log p_t(x)$ is known analytically. In addition, the PF-ODE vector field is well defined everywhere, so we can evaluate the corresponding Fisher speed for any $(\boldsymbol{x}_t, t)$ in spacetime to obtain the heatmaps.

For Figure 3, we compute the curvature using Christoffel symbols. Gaussian mixture prior $h_0 = \sum_{i=1}^{3} \pi_i \mathcal{N}(\mu_i, \sigma^2)$ with $\mu_1 = -1$, $\mu_1 = 0$, $\mu_3 = 1$, $\pi_1 = 1/3$, $\pi_2 = 1/3$, $\pi_3 = 1/3$ and $\sigma = 0.01$. We use the sigmoid schedule presented above. Figure 5 shows the corresponding spacetime curvature for the other noise schedules considered in this paper.

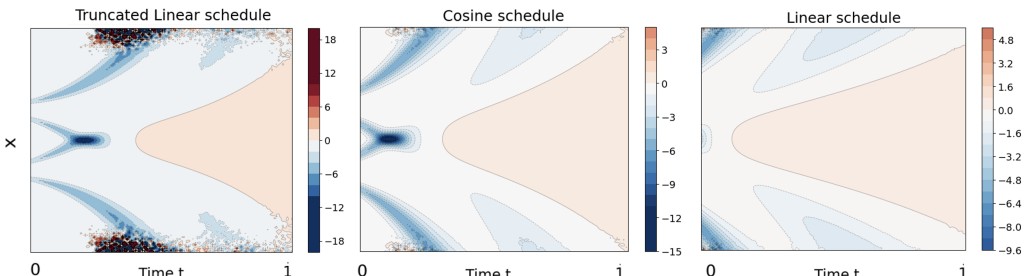

Figure 5: Sectional curvature of the spacetime manifold for different noise schedules.

As a sanity check, we calculated the curvature of spacetime for a 1D gaussian distribution prior. In Figure 6 we can see that the curvature is indeed constant negative, as expected from Lemma 10.

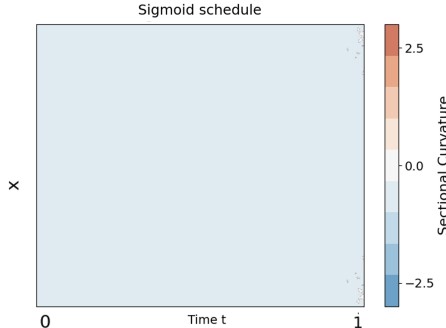

Figure 6: Sectional curvature of the spacetime manifold for different noise schedules.

