# OpenReview forum: "On the Fisher Geometry of Diffusion Models' Latent Space"
_ICLR.cc/2026/Workshop/GRaM — ICLR 2026 Workshop GRaM Poster_

### Official Review · Reviewer_6jYE · 2026-02-19
**Promising Geometric Perspective on Noise Schedules, but Requires Better Exposition**

**Rating:** 4
**Confidence:** 2

**Review:**

**Summary:**

This paper addresses the important problem of understanding how the choice of noise schedule impacts the performance of diffusion models. The authors address this by analyzing the latent space through the lens of Information Geometry, treating the generative process as a statistical manifold with a corresponding Fisher-Rao metric. They mathematically demonstrate that variance-preserving noise schedules are isometric (equivalent up to a time reparameterization). Furthermore, by assuming a Gaussian Mixture Model (GMM) prior, they decompose the Fisher metric to provide a geometric interpretation of the "phase transitions" that occur during denoising, linking sharp changes in the responsibility of mixture components to regions of negative sectional curvature. The authors argue that what makes a scheduler "bad" is not changing the shape of the manifold, but rather navigating this manifold in "wrong/high speed".

**Strengths:**

*1. Novel Perspective:* The connection between noise schedules, the Fisher-Rao metric, and curvature is a mathematically elegant way to formalize empirical observations about diffusion model training.
*2. Theoretical Utility:* Decomposing the Fisher metric into intra-cluster variance and a non-conservative phase-transfer term ($G(\theta)$) provides a principled way to define the "critical windows" or phase transitions in generative models.

**Weaknesses & Questions:**

While the core idea is strong, the paper in its current form is quite difficult to follow for an audience not simultaneously expert in both differential geometry and diffusion models. The presentation often obscures the main contributions.

*1. Missing Definitions and Intuition:* The paper introduces several critical concepts without sufficient grounding. For example, "Fisher Speed" is central to the analysis in Figure 2, but it is never formally defined in the text or appendix. Similarly, while "responsibility" is defined mathematically in Definition 3, the intuition for how it drives the geometry is rushed.

- Suggestion: Adding an appendix section dedicated to bridging the intuition between diffusion SDEs and Information Geometry (e.g., explicitly defining Fisher Speed and translating geometric distortion into ODE integration errors) would greatly improve accessibility.

*2. Structural Clarity and Answering the Core Question:* The paper clearly asks how different noise schedules impact the geometry. However, the structure makes it difficult to separate the mathematical setup from the actual answer. The paper proves that the manifolds are isometric, but the crucial conclusion—that practically fixing the coordinate chart causes numerical ODE solvers to struggle through regions of high curvature—is left somewhat implicit, if I understood this correctly. The connection between the theoretical isometry and the practical failure modes of certain schedules needs to be stated more explicitly.

*3. Justification of the "Three Phases" and Figure 3:* The qualitative division of the generative process into three phases feels somewhat disconnected from the math preceding it. Furthermore, it is unclear if Figure 3 is meant to be a schematic illustration or a plotted result from a specific system.

- Suggestion: The authors should explicitly state that Figure 3 is a schematic (if so) and more clearly tie the boundaries of the three phases back to the actual empirical observations of the GMM system shown in Figure 2 and the appendix.

**Minor Issues / Formatting:**

- *Citation Formatting:* There is inconsistent use of \citep and \citet throughout the text (e.g., lines 26/27, 179-181, 200, 559). Please ensure parenthetical vs. inline citations are used correctly.
- *Equation Numbering:* The main text stops numbering equations after Equation (2). Several important equations are unnumbered, making them difficult to reference. Numbering resumes later in the appendix; this should be made consistent.

**Pmlr Suitability:**

NA

---

### Official Review · Reviewer_SVVv · 2026-02-24
**Interesting analysis fit for a tiny paper**

**Rating:** 7
**Confidence:** 2

**Review:**

# Summary
The paper presents an analysis of the denoising process through the lens of the curvature of the associated statistical manifold. In a simple Gaussian example, they relate curvature to a change in responsibility and thus identify a shift toward negative curvature as an indicator of phase transitions. They demonstrate their results in illustrative simulations.

# Quality
+ The paper is easy to read and seems to point to relevant and recent related work. The illustrations help visualize the results.
* I did not check the proofs.
# Originality
+ I am not an expert, but the connection made in the paper does not seem to be worked out in the most prominently cited papers.

# Significance
- The paper sheds light on the effect of noise schedules from an information-theoretic perspective in a *simplified setting*, but provides little perspective if or how the insights might generalize to complex data distributions. In particular, it does not provide guidance on how the phenomenon might be useful in *choosing or constructing noise schedules*.

# Comments
* The paper often cites (Stancevic et al., 2025), it would be nice to compare the entropic schedules derived in (Stancevic et al., 2025) in Figure 2, in order to judge the effect of their optimal schedules.
* l. 584f I don't understand the implications of the statement. Please explain it in more detail in the final version.

**Pmlr Suitability:**

NA

---

### Meta-Review · Area_Chair_9CKB · 2026-02-24

**Decision:**

Accept

**Metareview:**

The reviewers claim the paper presents a novel and strong core idea, therefore I recommend acceptance. However, the reviewers also point out weaknesses in the presentation of the paper making hard to follow for a more general audience. Perhaps the authors can take some of this feedback into account to improve the paper.

**Relevance To Proceedings:**

Tiny paper — does not apply

**Relevance To Workshop:**

Yes — suitable for GRaM

---

### Decision · Program_Chairs · 2026-03-02

Accept (Poster)